# Non-Typical Clinical Presentation of COVID-19 Patients in Association with Disease Severity and Length of Hospital Stay

**DOI:** 10.3390/jpm13010132

**Published:** 2023-01-10

**Authors:** Alexandros Skourtis, Konstantinos Ekmektzoglou, Theodoros Xanthos, Stella Stouraitou, Nicoletta Iacovidou

**Affiliations:** 1Postgraduate Study Program (MSc) “Resuscitation”, School of Medicine, National and Kapodistrian University of Athens, 115 27 Athens, Greece; 2Department of Emergency Medicine, “Evangelismos” General Hospital, 106 76 Athens, Greece; 3School of Medicine, European University Cyprus, Nicosia 2404, Cyprus; 4Department of Midwifery, School of Health and Care Sciences, University of West Attica, 122 43 Aigaleo, Greece; 5Department of Neonatology, School of Medicine, Aretaieio Hospital, National and Kapodistrian University of Athens, 115 28 Athens, Greece

**Keywords:** COVID-19 symptoms, atypical presentation, length of stay, ICU, emergency department

## Abstract

Background: This study aimed to investigate the incidence of non-typical symptoms in ambulatory patients with mild-to-moderate COVID-19 infection and their potential association with disease progression. Materials and methods: Data on the symptomatology of COVID-19 patients presenting to the fast-track emergency department were collected between March 2020 and March 2021. Fever, cough, shortness of breath, and fatigue-weakness were defined as “typical” symptoms, whereas all other symptoms such as nasal congestion, rhinorrhea, gastrointestinal symptoms, etc., were defined as “non-typical”. Results: A total of 570 COVID-19 patients with a mean age of 42.25 years were included, the majority of whom were male (61.3%; N = 349), and were divided according to their symptoms into two groups. The mean length of hospital stay was found to be 9.5 days. A higher proportion of patients without non-typical symptoms were admitted to the hospital (*p* = 0.001) and the ICU (*p* = 0.048) as well. No significant differences were observed between non-typical symptoms and outcome (*p* = 0.685). Patients who did not demonstrate at least one non-typical symptom had an extended length of stay (*p* = 0.041). No statistically significant differences in length of hospital stay were associated with individual symptoms. Conclusion: With the possible exception of gastrointestinal symptoms, non-typical symptoms of COVID-19 at baseline appear to predispose to a milder disease.

## 1. Introduction

At the end of 2019, Wuhan Municipality in Hubei Province, China, reported a cluster of pneumonia cases with unknown etiology, followed by a global pandemic. Chinese authorities identified a new type of coronavirus (novel coronavirus, 2019-nCoV) as the causative agent of this outbreak, which was isolated on 7 January 2020. In February 2020, the World Health Organization (WHO) named the disease COVID-19, short for “coronavirus disease 2019” [1]. The incidence of non-typical disease symptoms and their association with severe disease were not well defined at the time. The most common symptoms of the disease (both in in- and outpatient settings), especially as seen during the first two waves of the pandemic, were fever, cough (with or without shortness of breath), fatigue, and, less commonly, sore throat, diarrhea, nausea/vomiting, headache, loss of taste or smell, rhinorrhea, abdominal pain, and rash [2,3,4,5,6,7,8,9].

Comorbidities (i.e obesity), older age, lymphocytopenia, and higher lactate dehydrogenase at presentation were found to be independent high-risk factors for COVID-19 progression [10,11]. Gradually, various predictive scores (4C Mortality Score, 4C Deterioration score) were developed, incorporating clinical and laboratory variables for prognostic purposes [12,13]. Symptoms of COVID-19 were utilized, even reported through smartphone applications, to predict probable infection [14] or clusters of symptoms [15], identify COVID-19 [16], or distinguish it from other respiratory viruses. [17]

This study aims to investigate the incidence of non-typical symptomatology in ambulatory patients during the early pandemic waves and their potential association with disease severity, i.e., if they required hospital or Intensive Care Unit (ICU) admission and mortality.

## 2. Materials and Methods

This is a retrospective cohort study including a total of 570 ambulatory patients with mild-to-moderate COVID-19 infection who presented to the fast track department of the Emergency Room of the General Hospital of Athens “Evangelismos” between March 2020 and March 2021. The data for the study participants were retrieved from hospital records (the emergency department’s registration logbook and the hospital’s electronic medical records). Ethical clearance was obtained from the Institutional Ethics Committee.

Fever, cough, shortness of breath, and fatigue-weakness were defined as “usual” or “typical” symptoms, whereas symptoms such as nasal congestion and rhinorrhea, sore throat, chest pain, smell and taste disorders, headache, myalgia, gastrointestinal symptoms (diarrhea nausea, vomiting, abdominal pain), rash, or others (arthralgia, syncope, palpitations, vertigo, anorexia, conjunctivitis, flank pain, hematuria-dysuria, earache, hoarseness, xerostomia) were defined as “non-typical”. Patients were divided into two groups based on the symptoms reported.

Inclusion criteria:Age between 15 and 85 years old.Saturation of Oxygen above 94% in room air (FiO2:21%).Symptoms including headache, smell and taste disorders, abdominal pain, diarrhea, vomiting, myalgia, or common respiratory infection symptoms (fever, cough, shortness of breath, fatigue).Respiratory Rate (R-R) < 20 per minute.Positive nasopharyngeal swab test for SARS-CoV-2 (RT-PCR or POC-PCR).Exclusion criteria:Age <15 years old or >85 years old.Oxygen saturation (SatO2) <94% in room air.Respiratory Rate RR > 20.Negative nasopharyngeal swab test for SARS-CoV-2 (RT-PCR or POC-PCR).Asymptomatic patients.

Factors that were evaluated included hospitalization, the need to admit to the intensive care unit (ICU), and the duration of their hospitalization until death, discharge, or referral to another hospital.

### Statistical Analysis

Testing of the variant normal distribution hypothesis was performed using the Kolmogorov–Smirnov tests for samples of over 30 experimental units and Shapiro–Wilk, for samples of less than 30 experimental units. The Pearson’s X^2^ test was used to check the association between two qualitative variables. The non-parametric Mann–Whitney test was used to check the equality of the mean values of two samples, which does not necessitate the normality of the data. Survival curves were plotted using the Kaplan–Meier survival curve to compare the probability of discharge at different points of time and to compare between two groups. Log-rank test was applied to find the statistical significance of the mean length of stay. A multivariate logistic regression was performed to assess the relation between Admission and the explanatory variables: Fever, Cough, Shortness of Breath, Fatigue, Nasal Congestion/Rhinorrhea, Sore Throat, Chest Pain, Smell and Taste Disorders, Headache, Myalgia, GI Disorders, and Other. Data were checked for multicollinearity with the Belsley-Kuh-Welsch technique. Heteroskedasticity and normality of residuals were assessed respectively by the White test and the Shapiro–Wilk test. Data were analyzed using the software IBM SPSS Statistics version 25.0. Statistical significance was considered if the *p*-value was <0.05.

## 3. Results

Of the 570 participants, the majority were male (61.3%, N = 349) with a mean age of 42.25 years ($\tilde{x}$ = 41, IQR = 25). All patients were unvaccinated, and there were no previous infections with Sars-Cov-2 reported or registered. The most common symptoms were fever (76.3%; N = 435) and cough (29.1%; N = 166), and other symptoms appeared at much lower rates (Table 1), (Figure 1).

The majority of patients (87.5%; Ν = 499) had “typical” symptoms, whereas 50.2% (Ν = 286) presented with at least one of the “non-typical” symptoms. The first group (n = 284) had no “non-typical“ symptoms, and the second group (n = 286) had solely “non-typical“ symptoms or “non-typical“ in combination with “typical”.

A total of 33% (N = 188) of patients required hospitalization and 2.3% (N = 13) were admitted in an ICU. The average length of stay in the hospital, among the hospitalized patients, was 9.5 days ($\tilde{x}$ = 7, IQR = 8). Finally, 0.9% (N = 5) died.

The association between “non-typical“ symptoms and hospitalization, death, and ICU admission is outlined in Table 2. A statistically significant (X^2^ = 10.67, *p* = 0.001) higher proportion of patients without non-typical symptoms (39.4% vs. 26.6%) were hospitalized. However, there were no differences in mortality (X^2^ = 0.209, *p* = 0.685) between patients with “non-typical” symptoms versus those without (0.7% vs. 1.1%). In contrast, there seemed to be statistically significant differences in ICU admission (X^2^ = 3.908, *p* = 0.048), with a higher proportion of patients admitted to the ICU (3.5% vs. 1%).

Table 3 shows the results of an X^2^ test to evaluate the incidence of symptoms related to hospitalization. Statistically significant differences (X^2^ = 124.234, *p* < 0.001) were found with the majority of patients experiencing shortness of breath (55.2%), fatigue (51.4%), cough (48.8%), chest pain (48.6%), diarrhea/nausea/vomiting/abdominal pain (44.4%), and fever (39.8%).

In multivariate analysis, GI Disorders (OR = 2.05, [1.13; 3.72], *p* = 0.0176), Shortness of Breath (OR = 2.64, [1.05; 6.62], *p* = 0.0384), Cough (OR = 2.82, [1.85; 4.3], *p =* 0.0001), Fatigue (OR = 3.09, [1.71; 5.58], *p* = 0.0002), and Fever (OR = 6.64, [3.47; 12.7], *p* 0.0001) were associated with higher rates of Admission. Nasal Congestion/Rhinorrhea (OR = 0.42, [0.12; 1.48], *p* = 0.1749), Sore Throat (OR = 0.47, [0.19; 1.17], *p* = 0.1047), Myalgia (OR = 0.48, [0.21; 1.09], *p* = 0.0795), Smell and Taste Disorders (OR = 0.56, [0.23; 1.35], *p* = 0.1962), Headache (OR = 0.82, [0.38; 1.79], *p* = 0.6183), Other (OR = 1.57, [0.64; 3.86], *p* = 0.3243), and Chest Pain (OR = 2.1, [0.98; 4.51], *p* = 0.0571) were not associated with the rate of Admission (Table 4), (Figure 2).

For the subset of the 188 hospitalized patients ($\bar{x}$
= 51.24, $\tilde{x}$ = 53 years, s = 14, 66% male and 34% female), we investigated whether the length of stay depended on the rarity of symptoms. Those who did not have at least one of the atypical symptoms were hospitalized for more days (*p* = 0.041). The Log Rank test (Mantel-Cox) demonstrated a statistically significant difference (*p* = 0.028) in the mean length of stay between the two groups (11,330 days in the group of patients with only typical symptoms vs. 8321 days in the group with at least one atypical symptom) (Figure 3).

There was no statistically significant relationship revealed between individual symptoms upon presentation and length of stay (Table 5).

## 4. Discussion

The main focus of COVID-19 research today is to highlight more clearly the biochemical processes the human body undergoes when exposed to the virus and the pharmacological effects of various medications in preventing or curing the disease. At the onset of the pandemic, however, registration of symptomatology and laboratory findings played an important role in understanding the disease, with a vast clinical picture ranging from asymptomatic to severe fatal disease [8].

Of the 570 participants, the majority were male (61.3%, N = 349) with a mean age of 42.25 years. The mean age of the patients vastly varies between studies, as do the ratios of male-to-female patients. For instance, a Saudi Arabian observational cohort study of 440 patients, with a mean age of 38.16 ± 13.43 years ($\tilde{x}$ = 36.00) and 321 (80%) male cases, is the only study that, to our knowledge, deals with cases with mild-to-moderate COVID-19 disease, however, it is limited to hospitalized patients [18]. In contrast, in a study of 1099 patients in China, the median age was 47 years (interquartile range, 35 to 58), and 41.9% were female [19]. The discrepancies between these results and those of our study might have occurred mainly because of the selection of a mild-to-moderate ambulatory group of COVID-19 patients, which at least to our knowledge has not been researched to a great extent. Furthermore, the age range, ethnicity, as well as other specifics of each study population might well play a significant role in the fluctuation of demographic proportions across various research.

In a study of 13,203 COVID-19 hospitalizations in Bologna, Italy, during the period from 1 February 2020 to 10 May 2021, the median length of stay was six days [20]. In our analysis, the median length of stay (LoS) was 7 days, which is comparable with the value reported in similar studies [21,22,23] but shorter than one in a study on China [24]. However, LoS reported within the ISARIC report (which includes data from 25 countries, despite having the bulk of patients from the UK), [25] specifically provided a median of 4 days (IQR = 1–9). Differences in LoS duration between regions can be attributed to the different strategies or practices employed to manage COVID-19 infection or the different populations investigated.

Data in our study regarding the most frequent symptoms, which are often also referred to as typical symptoms (fever, cough, shortness of breath, and fatigue), especially at the onset of a pandemic, seem to agree with data from retrospective studies and reviews of the initial period of COVID-19 [3,26,27]. Differences were noted primarily in the incidence of gastrointestinal symptoms (11.1%), generally differentiated between studies [28] and by the age of the patients [3,27]. A review and meta-analysis of 5196 COVID-19 patients revealed GI symptoms such as nausea/vomiting and diarrhea to be reported in 5 and 6% respectively, significantly less prevalent than respiratory symptoms [29]. Furthermore, distinctions were observed in the proportion of shortness of breath (5,1%) which might be associated with the fact that we studied patients with mild or moderate COVID-19 disease, in which the presence of severe respiratory infection symptoms such as shortness of breath are absent [30].

In a multicenter prospective observational study of 60,109 symptomatic COVID-19 patients hospitalized in 43 countries during the early stages of the pandemic, “typical” symptoms of fever (69%), cough (68%), and dyspnea (66%) were reported more frequently, with 92% of patients reporting at least one of them [31]. The data appear to be in partial agreement with our research results. This might be possibly linked to the fact that the majority of hospitalized patients presented primarily with ‘typical’ symptoms of COVID-19, whereas the likelihood of “non-typical” symptoms coexisting with “typical” among patients treated, particularly those admitted to the hospital with mild or moderate illness, is not well documented in the current literature.

The prevalence of diarrhea or gastrointestinal symptoms, in conjunction with the severity of the disease and the need for hospitalization, is controversial, with studies claiming that the presence of gastrointestinal symptoms is associated with more severe disease [19,28,32], whereas other studies support that it has no effect on mortality or morbidity [33,34,35]. Moreover, the interpretation of the results becomes significantly more challenging if one considers the use of treatments known to cause diarrhea, such as colchicine [36], which was widely used with COVID-19 patients in Greece [37,38].

The current study finds no statistically significant correlation between ‘non-typical’ symptoms and mortality. The literature supports a non-correlation of symptoms with COVID-19-related mortality [39], as evidenced by our study in which a similar proportion of patients died regardless of symptom. In accordance with a thorough literature study [40], although the incidence of cough and dyspnea is evident in patients dying from COVID-19, other symptoms such as myalgia or headache appear to have a homogenous distribution among both the survivors and those who deceased. In addition, dyspnea and severity of symptoms at baseline were the most important clinical predictive markers of mortality among the symptoms in a retrospective study of 2184 confirmed COVID-19 patients in Nigeria [41]. The absence of such a correlation between dyspnea and death in our trial could be attributed to the latter referring to ambulatory, clinically non-severe patients.

Furthermore, dyspnea has been described bibliographically as the only predictive symptom for severe illness and ICU admission [42]. In a retrospective study of 4997 COVID-19 patients, it was reported that most patients admitted to the intensive care unit had dyspnea as the presenting symptom, with diarrhea being the most common symptom among patients without severe disease at their initial presentation. In contrast, other symptoms, such as nausea, vomiting, fever, cough, asthenia, myalgia, pharyngitis, runny nose, anosmia, ageusia, headache, and chest pain, did not vary significantly from those admitted to the general ward [43].

In a retrospective study of 952 hospitalized COVID-19 patients, those with moderate disease had more “non-typical” symptoms such as headache, nausea, or vomiting than patients with severe disease [43,44]. In addition, the latter had shortness of breath more frequently than patients with milder disease [44,45]. In our research, a higher proportion of patients admitted to the ICU had no “non-typical” symptoms. This finding does not appear to align with the current literature. The discrepancy may be due to different symptom registration at each center and different ICU admissions criteria per area (which also depends on ICU bed availability).

Among hospitalized patients, fever, cough, and dyspnea are the most common symptoms at presentation, as reported in the literature [27,44,46], with frequency variations across different countries. At the same time, fatigue was more prevalent among patients aged between 40 and 70 [27]. In addition to fatigue, other common symptoms in hospitalized patients included diarrhea, nausea-vomiting, headache, and myalgia [44].

A meta-analysis of 3326 confirmed COVID-19 cases in China indicated that if the presenting symptoms were abdominal pain, dyspnea, hemoptysis, anorexia, diarrhea, or fatigue, the likelihood of progression to more severe disease was reduced [47], and thus, the requirement for hospitalization. Our study results are partly consistent with these observations since hospitalized patients’ most common symptoms were fatigue, cough, dyspnea, gastrointestinal symptoms, and fever.

In contrast, the presence of chest pain is also frequent among those hospitalized. That might be explained by the wide range of characteristics of chest pain (pleural type, stabbing or throbbing pain, precordial discomfort, weight, burning, etc.) in addition to the variety of possible pathophysiological mechanisms and causes related to chest pain [48] such as pleural damage or myocardial injury [49]. Moreover, since, as already reported, there is a possibility of cardiovascular or thromboembolic complications among COVID-19 patients [6,50,51,52,53,54,55,56], the presence of chest discomfort emerges the need for hospitalization in order to investigate its cause vigorously or follow-up over time. Inversely, symptoms such as runny nose/nasal congestion, sore throat, anosmia/ageusia, headache, myalgia, rash, and others were less common among patients in need of hospitalization.

To our knowledge, there does not appear to be a bibliographical reference regarding the length of hospital stay of patients who presented with “non-typical” symptoms versus those who did not.

In terms of the correlation of individual symptoms with hospital stay duration, references primarily focus on fever as a factor impacting hospitalization duration [57,58], whereas dyspnea appears to be a predictive indicator of prolonged length of hospital stay in a study conducted by a major hospital in Ethiopia [21]. On the other hand, a study of 730 COVID-19 patients in a tertiary hospital in northern India demonstrated that none of the symptoms were statistically significantly associated with a more extended hospital stay [22]. This conclusion seems consistent with the data from our study. Differences in predictive factors among individual symptoms concerning the length of stay could be attributed to the different protocols used for both admission and discharge of patients across different centers.

## 5. Conclusions

In our study, an attempt was made to approximate the presence of factors that could predict the severity of COVID-19 with purely clinical criteria without considering laboratory findings. The electronic records of ambulatory patients with mild-to-moderate COVID-19 infection were used where only symptoms were recorded (typical and non-typical) without comorbidities or baseline laboratory values.

With the possible exception gastrointestinal symptoms, the non-typical symptoms of COVID-19 at baseline appear to predispose to a milder disease, in the sense of no need for hospitalization or shorter hospitalization. No association of specific symptoms with hospitalization, admission to the ICU, or length of stay in the hospital was displayed.

A cohort study that could record and analyze the patient’s demographics, their laboratory findings, and an analysis of therapeutic interventions such as antiviral therapy might help to extract more significant conclusions.

## 6. Limitations

Limitations of the study were several, including no follow-up data for the studied subjects. Moreover, we have not studied the impact of treatment modalities or existing comorbidities on the length of stay. Data were limited to our center alone, and in several cases, files were incomplete, not including patient demographics, especially in patients not admitted to the hospital; that resulted in the inability to integrate data into the statistical analysis factors such as age, gender, socio-economic status, and the presence of comorbidities. Even though “nοn-typical” symptoms are probably asked about much less frequently relative to “typical” symptoms during clinical history taking and symptoms like fever, cough, and dyspnea are elicited much more frequently because they are perceived to be of much greater clinical importance, we only used electronic records where all symptoms were asked for, so as to avoid bias.

## Figures and Tables

**Figure 1 jpm-13-00132-f001:**
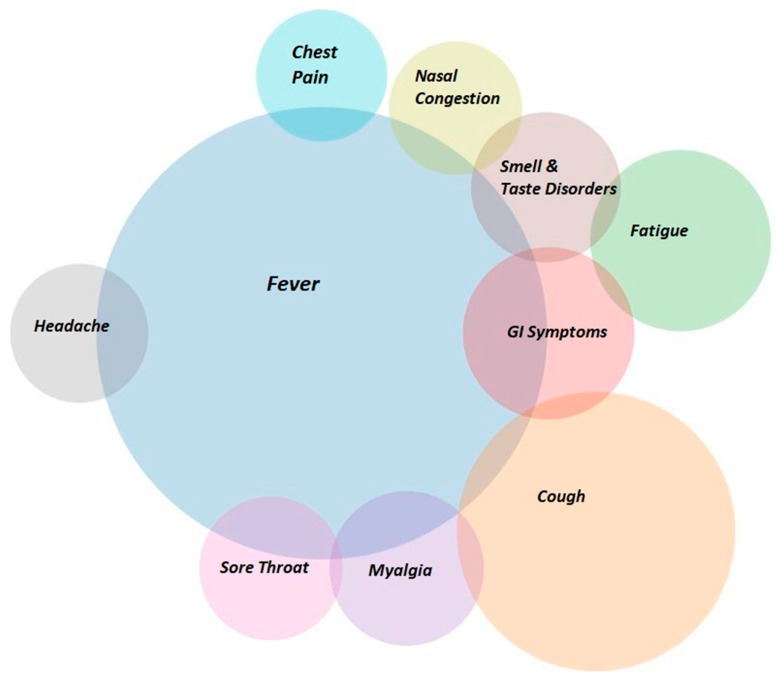
Venn Diagram of the most frequent symptoms.

**Figure 2 jpm-13-00132-f002:**
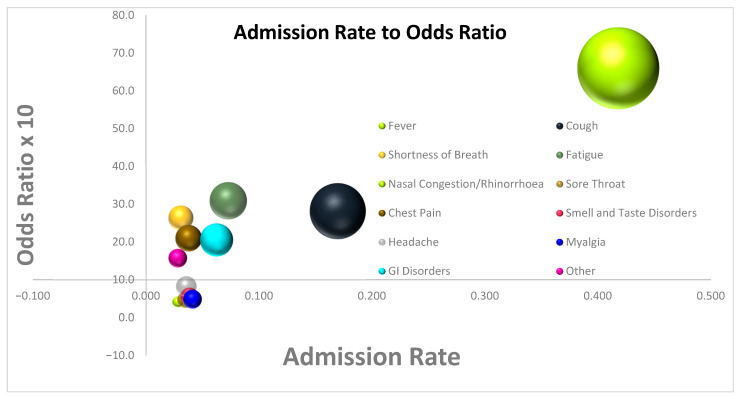
Bubble plot diagram of Admission Rate vs. Odds ratio for Admission for each of the symptoms.

**Figure 3 jpm-13-00132-f003:**
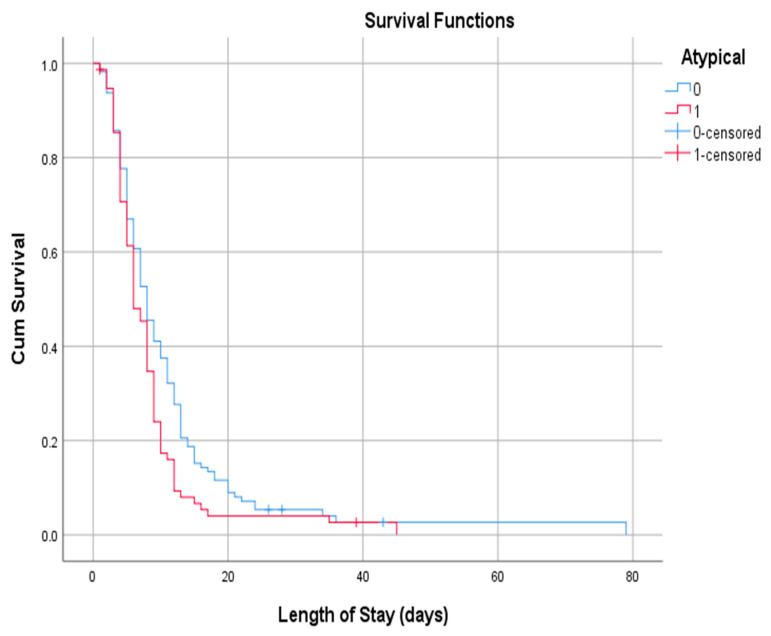
Kaplan–Meier Curves for Length of Stay.

**Table 1 jpm-13-00132-t001:** Demographic and clinical characteristics of participants.

	N	%
Male	349	61.3
Female	220	38.7
Age, $\bar{x}$ (standard deviation)	42.25 (15.37)	
Fever	435	76.3
Cough	166	29.1
Shortness of Breath	29	5.1
Fatigue	70	12.3
Nasal Congestion/Rhinorrhoea	38	6.7
Sore Throat	44	7.7
Chest Pain	37	6.5
Smell and Taste Disorders	48	8.4
Headache	41	7.2
Myalgia	51	8.9
GI Disorders	63	11.1
Rash	1	0.2
Other	32	5.6
Joint Pain	16	50.0
Hematuria-Dysuria	2	6.3
Heart Palpitations	1	3.1
Flank Pain	1	3.1
Anorexia	1	3.1
Impaired consciousness	1	3.1
Conjunctivitis	1	3.1
Dizziness and Vertigo	1	3.1
Hoarseness	1	3.1
Syncope	2	6.3
Xerostomia	1	3.1
Pre-syncope	3	9.4
Earache	1	3.1
Typical	499	87.5
Atypical	286	50.2
Admission	188	33.0
Death	5	0.9
ICU Admission	13	2.3
Length of Hospital Stay, $\bar{x}$ (standard deviation)	9.53 (8.99)	

**Table 2 jpm-13-00132-t002:** Association between “Non-typical” symptoms and patient hospital admission, Death, and ICU Admission.

		Non-Typical			
		No	Yes	X^2^	*p*
Admission	Yes	112 (39.4%)	76 (26.6%)	10.67	0.001
Death	Yes	3 (1.1%)	2 (0.7%)	0.209	0.685
ICU ^a^ Admission	Yes	10 (3.5%)	3 (1%)	3.908	0.048

^a^ ICU: Intensive Care Unit.

**Table 3 jpm-13-00132-t003:** X^2^ test results to evaluate the association of “Admission” variables and the presence of symptoms.

Admission				
	No	Yes	X^2^ 124.234	*p* < 0.001
**Fever**	262 (60.2%)	173 (39.8%)		
**Cough**	85 (51.2%)	81 (48.8%)		
**Shortness of breath**	13 (44.8%)	16 (55.2%)		
**Fatigue**	34 (48.6%)	36 (51.4%)		
**Nasal congestion and Rhinorrhea**	35 (92.1%)	3 (7.9%)		
**Sore Throat**	37 (84.1%)	7 (15.9%)		
**Chest pain**	19 (51.4%)	18 (48.6%)		
**Smell and taste disorders**	40 (83.3%)	8 (16.7%)		
**Headache**	30 (73.2%)	11 (26.8%)		
**Myalgia**	42 (82.4%)	9 (17.6%)		
**GI Symptoms ^a^**	35 (55.6%)	28 (44.4%)		
**Rash**	1 (100%)	0 (0%)		
**Other**	23 (71.9%)	9 (28.1%)		

^a^ GI: gastrointestinal.

**Table 4 jpm-13-00132-t004:** Multivariate Logistic Regression Symptoms to Admission.

Variable	Odds Ratio	*p*-Value
Fever	6.64 [3.47;12.7]	0.0001
Cough	2.82 [1.85;4.3]	0.0001
Shortness of Breath	2.64 [1.05;6.62]	0.0384
Fatigue	3.09 [1.71;5.58]	0.000176
Nasal Congestion/Rhinorrhea	0.415 [0.117;1.48]	0.175
Sore Throat	0.466 [0.185;1.17]	0.105
Chest Pain	2.1 [0.978;4.51]	0.0571
Smell and Taste Disorders	0.558 [0.231;1.35]	0.196
Headache	0.821 [0.377;1.79]	0.618
Myalgia	0.484 [0.215;1.09]	0.0795
GI Disorders	2.05 [1.13;3.72]	0.0176
Other	1.57 [0.64;3.86]	0.324

**Table 5 jpm-13-00132-t005:** Individual symptoms upon presentation and length of stay.

	Fever			
	No (N = 15)	Yes (N = 173)	Mann–Whitney	*p*
Length of Stay (days)	7 (3–9)	7 (5–12)	−1.294	0.196
	Cough			
	No (N = 107)	Yes (N = 81)	Mann–Whitney	*p*
Length of Stay (days)	7 (5–12)	8 (4–12)	−0.319	0.750
	Shortness of Breath			
	No (N = 172)	Yes (N = 16)	Mann–Whitney	*p*
Length of Stay (days)	7 (4.25–12)	8 (4–14.5)	−0.043	0.965
	Fatigue			
	No (N = 152)	Yes (N = 36)	Mann–Whitney	*p*
Length of Stay (days)	8 (4.25–12)	7 (4–12)	−0.372	0.710
	Sore throat			
	No (N = 181)	Yes (N = 7)	Mann–Whitney	*p*
Length of Stay (days)	7 (4.5–12)	8 (3–12)	−0.500	0.617
	Chest Pain			
	No (N = 170)	Yes (N = 18)	Mann–Whitney	*p*
Length of Stay (days)	8 (5–12)	5.5 (4–8.25)	−1.420	0.155
	Smell and Taste Disorders			
	No (N = 180)	Yes (N = 8)	Mann–Whitney	*p*
Length of Stay (days)	7 (4–12)	9.5 (8–11.5)	−1.082	0.279
	Headache			
	No (N = 177)	Yes (N = 11)	Mann–Whitney	*p*
Length of Stay (days)	7 (4–12)	7 (6–11)	−0.195	0.846
	Myalgia			
	No (N = 179)	Yes (N = 9)	Mann–Whitney	*p*
Length of Stay (days)	7 (4–12)	6 (4–14.5)	−0.072	0.942
	GI symptoms			
	No (N = 160)	Yes (N = 28)	Mann–Whitney	*p*
Length of Stay (days)	8 (4.25–12)	6.5 (4–9.75)	−0.985	0.325
	Other			
	No (N = 179)	Yes (N = 9)	Mann–Whitney	*p*
Length of Stay (days)	7 (4–12)	9 (5.5–9)	−0.334	0.739

## Data Availability

Data sharing is not applicable to this article.

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
