# Peer review of "Non-Typical Clinical Presentation of COVID-19 Patients in Association with Disease Severity and Length of Hospital Stay"

_jpm, 2023, doi:10.3390/jpm13010132_

Round 1
Reviewer 1 Report
comments for author
In the abstract “the majority 22 of whom were male (61.3%; N=349) with a mean age of 42.25 years” means that the age of male was 42.25, but it is the age of total patients. Please rewrite this sentence.
The conclusion in the abstract dose not represents your findings. Please re write them to emphasize the importance of not having atypical symptoms in covid-19 infected patients
It is not clear why patients with RR>20 and oxygen saturation of <94% were excluded? Pleas add a sentence as your logic
Author defines the study group as: (The first group (n=284) 87 had no "atypical" symptoms, and the second group (n=286) had solely "atypical" symptoms or "atypical" in combination with "typical"). In my opinion to conclude that the atypical symptom may have prognostic value (protective effect) , the typical symptoms should be the same between two groups. Did the author matched the typical ones between two groups?
For table 3, I think each row needs a p value
Author Response
It is with great pleasure to receive your comments regarding our manuscript entitled “Non-typical clinical presentation of Covid-19 patients in association with disease severity and length of hospital stay”.
Please find bellow a detailed answer to the comments raised. So as to assist the review process, Reviewer’s comments are highlighted in bold with our corresponding response being highlighted in italics.
Also, please note that a marked-up version with line numbering and all changes visible (e.g. highlighted in red) has been sent.
In the abstract “the majority 22 of whom were male (61.3%; N=349) with a mean age of 42.25 years” means that the age of male was 42.25, but it is the age of total patients. Please rewrite this sentence.
The abstract has been rewritten as recommended. It now reads: “Results: A total of 570 COVID-19 patients with a mean age of 42.25 years were included, the majority of whom were male (61.3%; N=349), and were divided according to their symptoms into two groups.”
The conclusion in the abstract dose not represents your findings. Please re write them to emphasize the importance of not having atypical symptoms in covid-19 infected patients.
We would like to thank the distinguished Reviewer for his/hers remark. The conclusion part of the Abstract has been rewritten as requested. The Conclusion section now reads “Conclusion: With the possible exception of gastrointestinal symptoms, the non-typical symptoms of covid-19 at baseline appear to predispose to a milder disease.”
It is not clear why patients with RR>20 and oxygen saturation of <94% were excluded? Please add a sentence as your logic
As clearly stated in the abstract, the aim of the study was to investigate the incidence of less frequent symptomatology in ambulatory patients with mild to moderate COVID-19 infection during the early pandemic waves and their potential association with disease severity. Therefore, we excluded patients with characteristics indicative of more severe disease (RR>20 and SpO2<94%). To further emphasize the above the first paragraph of the Materials and Methods section now reads: “It is a retrospective case series including a total of 570 ambulatory patients with mild to moderate COVID-19 infection that presented to the fast track department of the Emergency Room of the General Hospital of Athens “Evangelismos” between March 2020 and March 2021.”
Author defines the study group as: (The first group (n=284) 87 had no "atypical" symptoms, and the second group (n=286) had solely "atypical" symptoms or "atypical" in combination with "typical"). In my opinion to conclude that the atypical symptom may have prognostic value (protective effect) , the typical symptoms should be the same between two groups. Did the author matched the typical ones between two groups?
The Reviewer is absolutely spot on when making this distinction. However, as during data processing, we matched typical ones between the 2 groups and no correlation was recorded, we chose to compare typical vs non typical + typical symptoms. Besides, since the likelihood of "non-typical" symptoms coexisting with "typical" among patients treated, particularly those admitted to the hospital with mild or moderate illness, is not well documented in the current literature, we chose the aforementioned grouping scheme.
For table 3, I think each row needs a p value
A single X2 test was performed, assuming all positive ("yes") responses for all symptoms. It is appropriate to have this kind of presentation because, in most cases, we recorded multiple symptoms from multiple patients, and this is one approach to configure which symptom was the most frequent among admitted patients while comparing at the same all the other symptoms.
Reviewer 2 Report
This study is presented as a retrospective case series examining the association between COVID symptoms and disease severity as measured by hospitalization, LOS, ICU admission and mortality. Per the authors, the novelty of this study is its description of an "ambulatory" population, but their definition of this population is unclear as all patients presented to the ER and some were admitted to the hospital. There is also a distinction made between more frequently occurring and less frequently occurring symptoms. It is unclear to me why how commonly symptoms occur should have any bearing on disease severity and this is not adequately described by the authors. Ultimately they conclude that patients presenting with less frequently occurring symptoms (which also happen to be milder) are less likely to be admitted to the hospital or die.
Introduction
36 – not “resulting in,” perhaps you mean “followed by?”
40 – if the incidence of certain symptoms is not known, then they can’t be labelled “less typical.” Consider rewording this sentence for clarity
41-45 – does this symptom profile reflect patients seen and treated in an inpatient setting? If the references you cite have a mix of patient types, this should be clarified.
46 – the terms “less frequent,” “atypical” and “non-typical” are used interchangeably. Please use only one term and define it clearly.
Materials and Methods
This section needs a definition of outcomes being studies and description of statistical analysis.
51 – Would say “this is a retrospective” not “it is a retrospective”
51 – is this really just a case series? You are drawing conclusions regarding impact on mortality, etc… It seems like this would be more accurately described as a retrospective cohort study.
64-77 – Why were patients with hypoxia and tachypnea excluded? It seems that this exclusion would have a huge impact on outcomes. If you are trying to look only at “ambulatory patients” did you only focus on the patients that were discharged after their ER assessment and then readmitted later?
Results
It is unclear to me whether the “atypical” symptom group consisted predominantly of patients who ONLY presented with atypical symptoms vs patients with mostly typical symptoms and only 1 or 2 atypical symptoms.
96 – do you mean there was no difference in mortality between patients with atypical symptoms vs those without? Please reword this in a clearer was and provide mortality values in parentheses.
Table 2 – the number of patients WITHOUT each studies outcome do not need by displayed
Table 3 – is the p-value for each of these symptoms as they relate to rate of admission <0.001? This doesn’t seem possible as there are some symptoms such as cough and fatigue where the percentages in each group are almost identical.
Line 48-49 of the introduction suggests that intubation would be investigated as a marker of severity, but was not mentioned in the results section.
Discussion
Most of the discussion summarized data that is already known about COVID symptomatology and its relationship with disease severity. There is no mention of how or why the type of symptoms you are discussing (atypical vs typical) have any bearing of hospitalization, mortality, etc… Are you saying that atypical symptoms are not clinically important relative to the more severe “typical” symptoms or are you saying that the presence of so-called atypical symptoms is somehow protective?
143 – it is difficult to compare your admitted cohort to others because you excluded sicker patients with hypoxemia and tachypnea.
201-202 – you say that your findings do not align with literature and then say that they do “to a certain extent.” Can you clarify this?
226 – you say there are no studies looking at atypical symptoms and hospital LOS, but the next paragraph cites several. Can you clarify this?
Conclusions
240-242 – This is too broad a statement. You need to specifically mention the population you are studying and what factors you are trying to predict.
243 – what are “thoracic” symptoms exactly
243-246 – if no specific symptoms are associated with differences in disease severity, why do you single out “thoracic and gastrointenstinal” symptoms
Limitations
Limitations generally fall within the discussion section but this can be left to the discretion of the editors.
“Atypical” symptoms are probably asked about much less frequently relative to “typical” symptoms during clinical history taking. Symptoms like fever, cough and dyspnea are elicited much more frequently because they are perceived to be of much greater clinical importance. In a retrospective study, there is a huge bias towards underreporting of minor symptoms. Furthermore, the sicker a patient is, especially if there present with respiratory failure, it is very unlikely that the presence of “atypical” symptoms would be noted in the chart as the patient would be unable to provide a history.
Author Response
It is with great pleasure to receive your comments regarding our manuscript entitled “Non-typical clinical presentation of Covid-19 patients in association with disease severity and length of hospital stay”.
Please find bellow a detailed answer to the comments raised. So as to assist the review process, Reviewer’s comments are highlighted in bold with our corresponding response being highlighted in italics.
Also, please note that a marked-up version with line numbering and all changes visible (e.g. highlighted in red) has been sent.
Comments and Suggestions for Authors
This study is presented as a retrospective case series examining the association between COVID symptoms and disease severity as measured by hospitalization, LOS, ICU admission and mortality. Per the authors, the novelty of this study is its description of an "ambulatory" population, but their definition of this population is unclear as all patients presented to the ER and some were admitted to the hospital. There is also a distinction made between more frequently occurring and less frequently occurring symptoms. It is unclear to me why how commonly symptoms occur should have any bearing on disease severity and this is not adequately described by the authors. Ultimately they conclude that patients presenting with less frequently occurring symptoms (which also happen to be milder) are less likely to be admitted to the hospital or die.
We would like to thank the distinguished Reviewer for his/her commentary. As clearly stated in our manuscript, the aim of the study was to investigate the incidence of less frequent symptomatology in ambulatory patients with mild to moderate COVID-19 infection during the early pandemic waves and their potential association with disease severity. At the onset of the pandemic, registration of symptomatology and laboratory findings played an important role in understanding the disease with a vast clinical picture ranging from asymptomatic to severe fatal disease. To our knowledge, there does not appear to be any published study regarding the length of hospital stay of patients who presented with 'atypical' symptoms versus those who did not during the first pandemic wave. Hopefully the above, followed by a point by point response to the comments raised, will ease the Reviewer’s objections.
Introduction
36 – not “resulting in,” perhaps you mean “followed by?”
“Resulting” has been replaced by “followed by”, as requested.
40 – if the incidence of certain symptoms is not known, then they can’t be labelled “less typical.” Consider rewording this sentence for clarity
The sentence has been rewritten, as requested. Instead of “The incidence of less typical disease symptoms and their association with severe disease were not well defined at the time”, it now reads “The incidence of disease symptoms and their association with severe disease were not well defined at the time”
41-45 – does this symptom profile reflect patients seen and treated in an inpatient setting? If the references you cite have a mix of patient types, this should be clarified.
The aforementioned sentence has been clarified, as requested. It now reads “The most common symptoms of the disease (both in in- and outpatient settings), especially as seen during the first two waves of the pandemic, were fever, cough (with or without shortness of breath), fatigue, and, less commonly, sore throat, diarrhea, nausea/vomiting, headache, loss of taste or smell, rhinorrhea, abdominal pain, rash”
46 – the terms “less frequent,” “atypical” and “non-typical” are used interchangeably. Please use only one term and define it clearly.
The “non-typical” term has been used throughout the manuscript replacing “less frequent” and “atypical” when necessary, as requested. All changes have been highlighted with red throughout the revised version of the manuscript.
Materials and Methods
This section needs a definition of outcomes being studies and description of statistical analysis.
We would like to thank the distinguished Reviewer for the most insightful remarks. Hopefully, the revisions pointed out below will help defining our study better (alongside a detailed statistical analysis paragraph).
51 – Would say “this is a retrospective” not “it is a retrospective”
Corrected, as requested.
51 – is this really just a case series? You are drawing conclusions regarding impact on mortality, etc… It seems like this would be more accurately described as a retrospective cohort study.
Corrected, as requested.
64-77 – Why were patients with hypoxia and tachypnea excluded? It seems that this exclusion would have a huge impact on outcomes. If you are trying to look only at “ambulatory patients” did you only focus on the patients that were discharged after their ER assessment and then readmitted later?
As clearly stated in our manuscript, the aim of the study was to investigate the incidence of less frequent symptomatology in ambulatory patients with mild to moderate COVID-19 infection during the early pandemic waves and their potential association with disease severity. Therefore, patients with severe respiratory infection symptoms with tachypnea, hypoxia were excluded. The study only focused in this subset of patients presented to the fast track department of the Emergency Room and were either admitted or discharged.
Results
It is unclear to me whether the “atypical” symptom group consisted predominantly of patients who ONLY presented with atypical symptoms vs patients with mostly typical symptoms and only 1 or 2 atypical symptoms.
Thank you for yours on the spot commentary. Since the likelihood of "non-typical" symptoms coexisting with "typical" among patients treated, particularly those admitted to the hospital with mild or moderate illness, is not well documented in the current literature, we chose the aforementioned grouping scheme. The first group had only "typical” symptoms, and the second group had solely "non-typical" symptoms or "non-typical" in combination with "typical”.
96 – do you mean there was no difference in mortality between patients with atypical symptoms vs those without? Please reword this in a clearer was and provide mortality values in parentheses.
The sentence “However, there were no significant differences (X2=0.209, p=0.685) between "non-typical" symptoms and death” was replaced by the sentence “However, there were no difference in mortality (X2=0.209, p=0.685) between patients with "non-typical" symptoms versus those without (0.7% vs 1.1%,)”.
Table 2 – the number of patients WITHOUT each studies outcome do not need by displayed
Revised, as requested.
Table 3 – is the p-value for each of these symptoms as they relate to rate of admission <0.001? This doesn’t seem possible as there are some symptoms such as cough and fatigue where the percentages in each group are almost identical.
A single X2 test was performed, assuming all positive ("yes") responses for all symptoms. It is appropriate to have this kind of presentation because, in most cases, we recorded multiple symptoms from multiple patients, and this is one approach to configure which symptom was the most frequent among admitted patients while comparing at the same time all the other symptoms.
Line 48-49 of the introduction suggests that intubation would be investigated as a marker of severity, but was not mentioned in the results section.
The Reviewer is absolutely right on his/her comment. The phrase “This study aims to investigate the incidence of non-typical symptomatology in ambulatory patients during the early pandemic waves and their potential association with disease severity, i.e., if they required admission to the hospital or possibly intubation during their stay in the hospital.” has been revised and now reads: “This study aims to investigate the incidence of non-typical symptomatology in ambulatory patients during the early pandemic waves and their potential association with disease severity, i.e., if they required hospital or Intensive Care Unit (ICU) admission and mortality”
Discussion
Most of the discussion summarized data that is already known about COVID symptomatology and its relationship with disease severity. There is no mention of how or why the type of symptoms you are discussing (atypical vs typical) have any bearing of hospitalization, mortality, etc… Are you saying that atypical symptoms are not clinically important relative to the more severe “typical” symptoms or are you saying that the presence of so-called atypical symptoms is somehow protective?
We would like to thank the distinguished Reviewer for the comment. Our data seem to point out that in ambulatory patients with mild to moderate COVID-19 infection during the first pandemic wave, the non-typical symptoms of covid-19 at baseline appear to predispose to a milder disease. This is clearly stated in the Abstract of our manuscript.
143 – it is difficult to compare your admitted cohort to others because you excluded sicker patients with hypoxemia and tachypnea.
We believe that this is one of the major advantages of the study, since there are no clinical data to our knowledge regarding this subset of patients published so far. Of course, the aforementioned comment by the Reviewer has scientific merit and is discussed in the final paragraph of the conclusion section in our manuscript where it reads that: “A cohort study that could record and analyze the patient's demographics, their laboratory findings, and an analysis of therapeutic interventions such as vaccination or antiviral therapy might help to extract more significant conclusions.”
201-202 – you say that your findings do not align with literature and then say that they do “to a certain extent.” Can you clarify this?
The phrase “to a certain extent” has been omitted.
226 – you say there are no studies looking at atypical symptoms and hospital LOS, but the next paragraph cites several. Can you clarify this?
The sentence “To our knowledge, there does not appear to be a bibliographical reference regarding the length of hospital stay of patients who presented with “non-typical” symptoms versus those who did not.” has been omitted.
Conclusions
240-242 – This is too broad a statement. You need to specifically mention the population you are studying and what factors you are trying to predict.
The sentence “The electronic records of ambulatory patients with mild to moderate COVID-19 infection were used where only symptoms were recorded (typical and non-typical) without comorbidities or baseline laboratory values.”
243 – what are “thoracic” symptoms exactly
See the comment below 243-246.
243-246 – if no specific symptoms are associated with differences in disease severity, why do you single out “thoracic and gastrointenstinal” symptoms
The phrase " thoracic symptoms" was omitted. In the multivariate logistic regression that was added in the revision, there seems to exist a statistical significant correlation between GI symptoms and a higher admission rate.
Limitations
Limitations generally fall within the discussion section but this can be left to the discretion of the editors.
“Atypical” symptoms are probably asked about much less frequently relative to “typical” symptoms during clinical history taking. Symptoms like fever, cough and dyspnea are elicited much more frequently because they are perceived to be of much greater clinical importance. In a retrospective study, there is a huge bias towards underreporting of minor symptoms. Furthermore, the sicker a patient is, especially if there present with respiratory failure, it is very unlikely that the presence of “atypical” symptoms would be noted in the chart as the patient would be unable to provide a history.
So as to address the Reviewer’s comment, the phrase “Even though “nοn-typical” symptoms are probably asked about much less frequently relative to “typical” symptoms during clinical history taking and symptoms like fever, cough and dyspnea are elicited much more frequently because they are perceived to be of much greater clinical importance, we only used electronic records where all symptoms were asked for, so as to avoid bias.” has been added in the Limitations section of our revised manuscript.
Reviewer 3 Report
Add SARS-CoV-2 variants circulating.
Add Co-morbidities.
Essential to know the biomarkers present, i.e., CRP, d-Dimer.
Add vaccination status.
Add previous infections with SARS-CoV-2.
Add multivariable logistic regression.
Use Venn diagrams.
Use bubble plots.
Add and review:
Huang C. Lancet 2020; 395:497.
Sudre CH. Science Advances 2021; 7:eabd4177.
Knight SR. BMJ 2020; 370:m3339
Gupta RK. Lancet Respir Med 2021; 9:349.
Dixon BE. Plos One 2021; 16:e0241875.
Pullen MF. Open Forum Infect Dis 2020; 7:ofaa271.
Kompaniyets L. MMWR; 70:355.
Manni C. Nat Med 2020; 26:1037.
Ji D. Clin Infect Dis 2020; 71:1393.
Jassat W. Lancet Glob Health 2021; 9:e1216.
Ghayda RA. Int J Environ Res Public Health 2020; 17:5026.
Ulicini M Life 2021; 11:561.
Author Response
It is with great pleasure to receive your comments regarding our manuscript entitled “Non-typical clinical presentation of Covid-19 patients in association with disease severity and length of hospital stay”.
Please find bellow a detailed answer to the comments raised. So as to assist the review process, Reviewer’s comments are highlighted in bold with our corresponding response being highlighted in italics.
Also, please note that a marked-up version with line numbering and all changes visible (e.g. highlighted in red) has been sent.
Add SARS-CoV-2 variants circulating.
According to the Greek National Public Health Organization
the variants of concern that were present in Greece during the duration of the trial besides the original variant were
B.1.1.7/UK lineage (Variant VOC_202012)
Variant with deletion in S Gene
8.1.351/South Africa (Variant 501V2)
VariantY453F
Variant19B/A.23.1
8.1.1.318
R.1
National Public Health Organization. Daily Epidemiological Surveillance Report of New Coronavirus Infections. Retrieved March 3, 2021, from https://eody.gov.gr/wp-content/uploads/2021/03/covid-gr-daily-report-20210304.pdf
Since the clinical picture associated with these variants did not significantly differ, we omitted the reference for straightforwardness.
Add Comorbidities.
Essential to know the biomarkers present, i.e., CRP, d-Dimer.
While it is essential to know comorbidities and biomarkers' values, such as CRP and d-Dimers, since we collected the data during the first two waves of the pandemic in Greece, hospital records and registries were incomplete to play a pivotal factor in any statistical process. Taking this in regard while highlighting it as a limitation to our study, we preferred a rather clinical approach to the study without taking comorbidities or lab testing (with the exception of PCR) into consideration.
Add vaccination status.
The vaccination status for all the subjects was 'unvaccinated.' It is added
Add previous infections with SARS-CoV-2.
No previous infections with Sars-Cov-2 were reported or registered for any of the subjects.
Add multivariable logistic regression.
The revised article's detailed statistical analysis paragraph will help elucidate the method used. A logistic regression analysis has been added.
Use Venn diagrams.
Use bubble plots.
Addressed as requested.
Add and review:
Huang C. Lancet 2020; 395:497.
It is now referenced in the introduction.
Sudre CH. Science Advances 2021; 7:eabd4177.
It is now referenced in the introduction.
Knight SR. BMJ 2020; 370:m3339
It is now referenced in the introduction.
Gupta RK. Lancet Respir Med 2021; 9:349.
It is now referenced in the introduction.
Dixon BE. Plos One 2021; 16:e0241875.
It is now referenced in the introduction.
Pullen MF. Open Forum Infect Dis 2020; 7:ofaa271.
It is now referenced in the introduction.
Kompaniyets L. MMWR; 70:355.
It is now referenced in the introduction.
Manni C. Nat Med 2020; 26:1037.
It is now referenced in the introduction.
Ji D. Clin Infect Dis 2020; 71:1393.
It is now referenced in the introduction.
Jassat W. Lancet Glob Health 2021; 9:e1216.
While that is a very interesting article, we believe that it wouldn’t be possible to be integrated into the conversation, especially with our current article data set, variables and results.
Ghayda RA. Int J Environ Res Public Health 2020; 17:5026.
It is now referenced in the discussion.
Ulicini M Life 2021; 11:561.
We are sorry, but we could not find the referred article.
Round 2
Reviewer 3 Report
Good work, thank you.